# Effective Crop Management and Modern Breeding Strategies to Ensure Higher Crop Productivity under Direct Seeded Rice Cultivation System: A Review

Nitika Sandhu [1], Shailesh Yadav [2] , Vikas Kumar Singh [3] and Arvind Kumar [2,4,*]

[1] School of Agricultural Biotechnology, Punjab Agricultural University, Ludhiana 141004, India; nitikasandhu@pau.edu
[2] IRRI South Asia Regional Centre (ISARC), Varanasi, Uttar Pradesh 221106, India; shaileshagri9@gmail.com
[3] International Rice Research Institute, South Asia Hub, ICRISAT, Patancheru 502319, India; v.k.singh@irri.org
[4] International Crops Research Institute for the Semi-Arid Tropics, Patancheru 502319, India
[*] Correspondence: Arvind.Kumar@cgiar.org

**Abstract:** Paddy production through conventional puddled system of rice cultivation (PTR) is becoming more and more unsustainable—economically and environmentally—as this method is highly resource intensive and these resources are increasingly becoming scarce, and consequently, expensive. The ongoing large-scale shift from puddled system of rice cultivation PTR to direct seeded rice (DSR) necessitates a convergence of breeding, agronomic and other approaches for its sustenance and harnessing natural resources and environmental benefits. Current DSR technology is largely based on agronomic interventions applied to the selected varieties of PTR. In DSR, poor crop establishment due to low germination, lack of DSR-adapted varieties, high weed-nematode incidences and micronutrient deficiency are primary constraints. The approach of this review paper is to discuss the existing evidences related to the DSR technologies. The review highlights a large number of conventionally/molecularly characterized strains amenable to rapid transfer and consolidation along with agronomic refinements, mechanization and water-nutrient-weed management strategies to develop a complete, ready to use DSR package. The review provides information on the traits, donors, genes/QTL needed for DSR and the available DSR-adapted breeding lines. Furthermore, the information is supplemented with a discussion on constrains and needed policies in scaling up the DSR adoption.

**Keywords:** direct seeded rice; genomics-assisted breeding; mechanization; nutrient; precision agronomy; QTL/genes; water; weed

## 1. Introduction

To meet the future food demand globally, the crop yield must be increased at an estimated annual growth rate of 2.4%. This hike is very much essential to double the food production by 2050 [1]. Rice is the most important staple food crop for almost half of the world's population. It is very important for the millions of farmers who grow rice on millions of hectares land, and even for the landless workers who derive their income from working on these farm lands [2]. Worldwide, Asia is the net rice exporter, accounting for ~70% of world rice exports. The traded volume of rice crop accounts for only around 7% of the total global consumption, which is very small. Globally, Asia is both the key supplier and consumer of rice crop. The trend of rice production in the world from 1994 to 2018 in the world (Figure 1A) and across regions (Figure 1B) are presented in Figure 1. Rice consumption is predicted to increase from 388 to 465 million tons from 2010 to 2035 in Asia [3]. An International Food Policy Research Institute report suggests a decline of 12–14% in the world's rice production by 2050 relative to 2000 because of climate change. Global rice production in Asia more than tripled from 1961 to 2010, with a compound

growth rate of 2.21% year$^{-1}$ [4]. The green revolution's success in 1960s witnessed per capita rice consumption rise from 85 to 100 kg from 1960 to 2010 [4]. Globally, at least 8 to 10 million tons of more rice per year with an annual increase of 0.6 t ha$^{-1}$ (1.2% to 1.5%) is needed in the coming decade to obtain the projected increase in rice production [4]. This has to be achieved from less land and fewer resources, and under adverse effects of climate change.

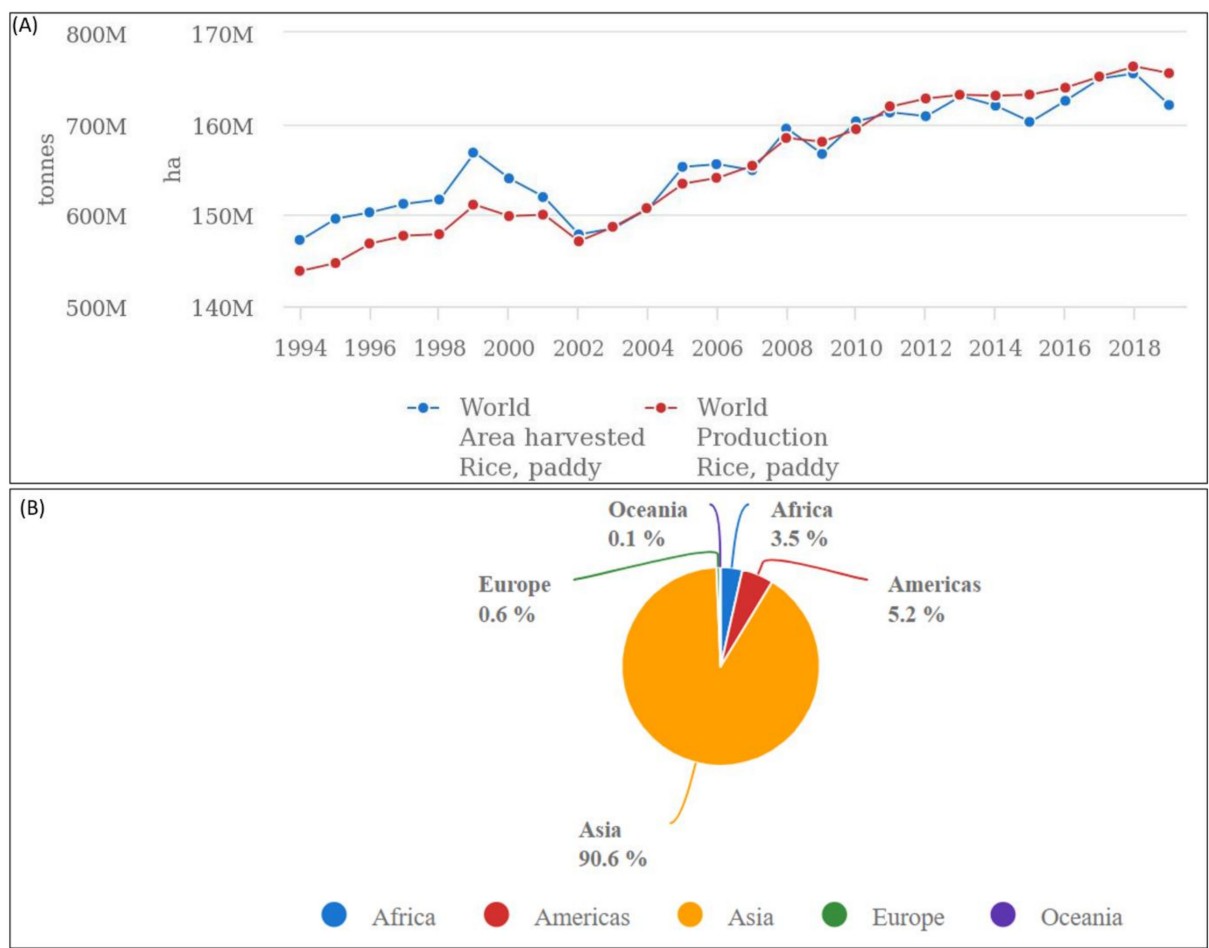

**Figure 1.** (**A**) Production/yield quantities of rice, paddy in world + (Total) from 1994–2019. (**B**) Production share of rice, paddy by region from 1994–2019. Source: FAOSTAT (http://www.fao.org/faostat/en/#data/QC/visualize accessed on 22 December 2020).

Rice contributes about two-thirds and one-third of the calorie intake for >3 billion people in Asia and >1.5 billion people in Africa and Latin America, respectively [5]. Irrigated rice shared 75% (410 million tons) of the world rice production per year [6]. Sustainable rice production is a key source of livelihood for about 140 million rice farming households and for the millions of poor rice farm laborers. A sufficient, affordable, and stable supply of rice strengthens the economic growth and political stability of the Asia-pacific region. A sustainable increase in rice production in the future to ensure a sufficient, affordable, and stable supply of rice to poor consumers remains an important challenge. Future rice farming faces several challenges, such as the urgent need to produce more rice to meet the world's rising demand from an increasing population, global climate change, environmental degradation linked with intensive cultivation practices, increasing competition for water-labor-land-energy, industrialization, and urbanization. Similarly, stability in the increasing prices of rice is also a key challenge in the context of biophysical, technological, socioeconomic, institutional, and policy constraints that is associated with the adoption of novel climate-smart varieties and practices.



Despite these challenges, several new high-level technologies for rice-based systems are available, which bring opportunities to increase rice production, and thereby, enhance food security and reduce poverty, hunger, and malnutrition. For example, rice yield can be increased using improved varieties, better agronomic practices, and better irrigation management. The labor input can be reduced using direct seeding, weed management, and mechanization. The environmental footprint and climate change can be decreased using innovative technologies such as direct-seeded rice cultivation practices; production cost can be reduced using resource-efficient technologies. In addition, innovative and mechanized postharvest technologies and practices related to drying, milling, and storage can reduce losses and increase product quality and the price premium for farmers.

Rice is produced in an extensive range of growing environment and locations such as from the wettest areas (Myanmar's Arakan Coast with an average annual rainfall of >5100 mm) to the driest deserts (Al Hasa Oasis in Saudi Arabia with average annual rainfall is <100 mm) [7]. Rice crop is generally grown by transplanting of seedlings into the puddled soil, which is a very water-labor-energy intensive system of cultivation [8]. The advantages of the PTR include weed suppression [9], creating anaerobic conditions to increase nutrient availability (iron, zinc, nitrogen, phosphorus) [10] and better seedling establishment. On the other hand, puddling leads to the higher water loss due to surface evaporation and water percolation [11], adversely affects the physical properties of soil [12], reduces permeability in the subsurface layers, and forms hard-pans at the shallow depths [13] which negatively affect the performance of succeeding crops [14]. Furthermore, puddling contributes to high-risk methane emissions [15,16]. It is also associated with farmers' practice of open field rice residue burning because of the short turnaround time between rice harvests and planting the next crop. The traditional system of cultivation necessitates transformational changes due to deteriorating water resources, insufficient labor, and the increasing labor prices. These factors call for a major shift from the traditional puddled transplanting system of rice cultivation to the direct seeding system of rice cultivation.

The direct-seeded rice (DSR) cultivation system involves sowing of seeds in the unsaturated and nonpuddled soil, in contrast to the traditional puddled transplanted system of rice cultivation (PTR), where the transplanting of seedling from the nursery to the puddled soil is required [17]. Currently, DSR is becoming very popular because it offers very exciting opening to improve water and environment sustainability [18,19]. To make DSR a successful technology, there is a strong need to develop early-maturing, short duration DSR-adapted rice varieties with complete site-specific water-nutrient-weed management package with increased adoption [20]. However, weeds, nematode infestation, increased incidences of biotic stresses, lodging, reduced grain quality and stagnant yields are the major problems associated with DSR. Bridging the existing yield gaps involving genetic improvement allow the strategic consequence on ensuring global food and nutrition security. DSR technology has been reported as a water-saving technology for both the rainfed and irrigated conditions, where at the farm level the availability of water is either too low or too costly for economic rice production [8]. In dry DSR, methods such as broadcasting, drilling, dibbling and direct seeding in dry soil using mechanization can be employed for the rice crop establishment [8]. In wet seeding, the pregerminated rice seedlings can be grown in the puddled rice field called as aerobic wet direct seeding or drilled into the puddled soil called as anaerobic wet direct seeding applying broadcasting or line sowing using a drum or anaerobic seeder [8,21]. In Asia, dry direct seeding is practiced in shallow lowland and rainfed upland areas [22], whereas wet seeding is generally practiced in eastern India and on a wide scale in Sri Lanka, Vietnam and Malaysia.

The adoption of DSR technology and water saving practices complemented by better infrastructure and market access will increase the income and livelihood of farmers and the poor. However, the irony is that, despite a big push from the suppliers and high demand from the consumers, adoption of DSR technology and practices is very low because of the problems associated with the technology, institutions, and policy. Large-scale dissemination and adoption of these high-level technologies for rice-based systems

can sustainably increase rice production, improve food security, reduce poverty, and accelerate rural transformation.

## 2. Benefits of DSR

### 2.1. Water and Labor Use

The agriculture water use is projected to rise by 20% in 2050, and the irrigation accounts for 70% of total global water withdrawals [23]. By 2025, the water availability for agriculture sector is expected to reduce by 10% [24]. By 2025, approximately 17 to 22 mha of the area under irrigated rice crop production is predicted to face severe water scarcity [25]. In Asia, the rice crop is responsible of the consumption of about 50% of the total available irrigation water, which accounts for 24% to 30% of the withdrawal of total freshwater globally [26,27]. The conventional PTR system of cultivation requires 25 to 50 person-days ha$^{-1}$ [28], while the size of agriculture workforce declined by approximately 30 million between 2004–2005 and 2011–2012 due to the rapid economic growth in non-agricultural sectors in the Asia [29] and the rising labor wages [30]. The puddling practices followed in the wetland rice production alone requires ~30% of the total crop water consumption. DSR have the great potential to save water and labor use compared to PTR.

In comparison to PTR, the water saving under DSR in the Philippines was 11% to 18% [31], in Malaysia 40% [32] and 10% to 50% was claimed in India [33–35]. DSR on raised beds reported 13% to 23% water savings compared to PTR, but this was also associated with 14% to 25% of yield reduction. Furthermore, the water use efficiency in the rice-wheat system was higher with DSR (0.45 g L$^{-1}$) than with PTR (0.37–0.43 g L$^{-1}$) [33]. Similarly, the labor requirement is lesser under DSR compared to PTR; the labor reduction ranged from 11% to 75% under DSR compared to PTR [36–43].

Case Study of Cambodia, Nepal and Punjab

In a study conducted in Cambodia under Asian Development bank funded project "Climate-smart practices and varieties for intensive rice-based systems in Bangladesh and Cambodia," water saving ranged from 19% to 32% was observed in different provinces of Cambodia under DSR compared to PTR (Table 1). Significant variation in the number of laborers used was observed in mechanized DSR and PTR, with an average reduction of 60–79% if shifting to mechanized DSR (Table 2). The labor savings of 43% to 49% (in terms of number of laborers, in person-days) were observed under DSR cultivation. The energy used for pumping water was significantly different under DSR compared to PTR (Table 3). The experiments conducted in Nepal reported 25% saving in total cost employing DSR in combination with mechanized insecticide/pesticide/herbicide spray and mechanized harvesting compared with the traditional PTR system. The DSR technology provided a reduction of approximately 83% total labor requirement compared to the PTR system. In a study conducted in Punjab, India, the DSR method of cultivation reported to save 3–4 irrigations and a 45% reduction in labor compared to PTR without any yield losses [44]. Consequently, DSR can help overcome the labor deficiency during the peak rice growing season.

**Table 1.** Water saving under mechanized DSR compared to PTR at Kampong Thom and Takeo sites, Cambodia.

| Province | District | PTR (m3) | Mechanized DSR (m3) | % Water Saving |
|---|---|---|---|---|
| Kampong Thom | Santuk | 1541.15 | 1181.15 | 23.36 |
| | Baray | 1101.15 | 751.06 | 31.79 |
| Takeo | Tram Kak | 1262.09 | 1561.19 | 19.16 |

**Table 2.** Labor saving under mechanized DSR compared to PTR at Kampong Thom and Takeo sites, Cambodia.

| Site | Traits | PTR | Mechanized DSR | Standard Deviation | Standard Error | *p*-Value |
| | | Mean | Mean | | | |
|------|--------|------|----------------|--------------------|----------------|-----------|
| Kampong Thom [†] | Number of labor (person) | 90 | 46.08 | 9.02 | 2.33 | 0.000 *** |
| | Number of day (hours) | 680 | 146.03 | 49.94 | 12.89 | 0.000 *** |
| | Cost (K Riel) | 1,635,000 | 1,233,857 | 130,238 | 33,627 | 0.000 *** |
| Takeo [ˣ] | Number of labor (person) | 100 | 56.72 | 20.05 | 6.34 | 0.000 *** |
| | Number of day (hours) | 325 | 140.13 | 51.42 | 16.26 | 0.000 *** |
| | Cost (K Riel) | 1,635,000 | 1,106,240 | 230,654 | 72,939 | 0.000 *** |

[†] Sample size: *n* = 15, [ˣ] Sample size: *n* = 10, *** statistical significance at 1%.

**Table 3.** Energy saving under mechanized DSR compared to PTR at Kampong Thom and Takeo sites, Cambodia.

| | Variable | PTR | Mechanized DSR | Standard Deviation | Standard Error | *p*-Value |
| | | Mean | Mean | | | |
|--|----------|------|----------------|--------------------|----------------|-----------|
| Kampong Thom | Average diesel (liter) used per growth stage | 30 | 16.11 | 4.62 | 2.67 | 0.035 *** |
| Takeo | | 25 | 18.92 | 12.32 | 5.03 | 0.28 ns |

*** significance at 1%, ns: not significant.

*2.2. Green-House Gas Emission*

In 2017, the global GHG emissions from the agricultural activities totaled about 582 million metric tons. The flooded puddled rice fields are the one of the most important sources of GHG emission. The anaerobic conditions in the PTR system led to the production of methane into the atmosphere through their roots and stems. Consequently, the water management and water saving technologies can help to reduce GHG emissions by ~90% as compared to the conventional PTR system of rice cultivation, while maintaining or improving the rice crop yield. India is the third main GHG emitter behind China and the United States. The agriculture and livestock sectors accounts for about 18% of the gross national GHG emissions. To feed the increasing world population with present dietary patterns, the overall food production is expected to hike by 70% between 2005–2050, which will further result in a 30% rise in global GHG emissions from the agriculture sector [45].

In a study conducted by Sapkota et al. [46], the results suggest that the crop contributed about 42% to the total agricultural emissions (Figure 2A). In India, the total GHG emissions from overall crop production were highest in Andhra Pradesh (AP), Uttar Pradesh (UP) and Maharashtra (MH), followed by West Bengal (WB), Madhya Pradesh (MP) and Punjab (Pb) (Figure 2B). The total GHG emissions from the paddy rice cultivation were reported highest in Andhra Pradesh followed by the West Bengal (WB), Assam (AS) and the Tamil Nadu (TN) (Figure 2C) and the emissions intensity for rice cultivation was at their peak in Himanchal Pradesh (HP) followed by Uttarakhand (UK), Kerala (KL), Assam (AS) and Karnataka (KA) (Figure 2D). Furthermore, this study showed that the GHG mitigation potential using less fertilizer and precise nutrient management was highest in Uttar Pradesh (UP) followed by Andhra Pradesh (AP), Maharasthra (MH) and Punjab (Pb) (Figure 2E). In addition, the proper water management practices led to the lowering of GHG emission in different states in India (Figure 2F).

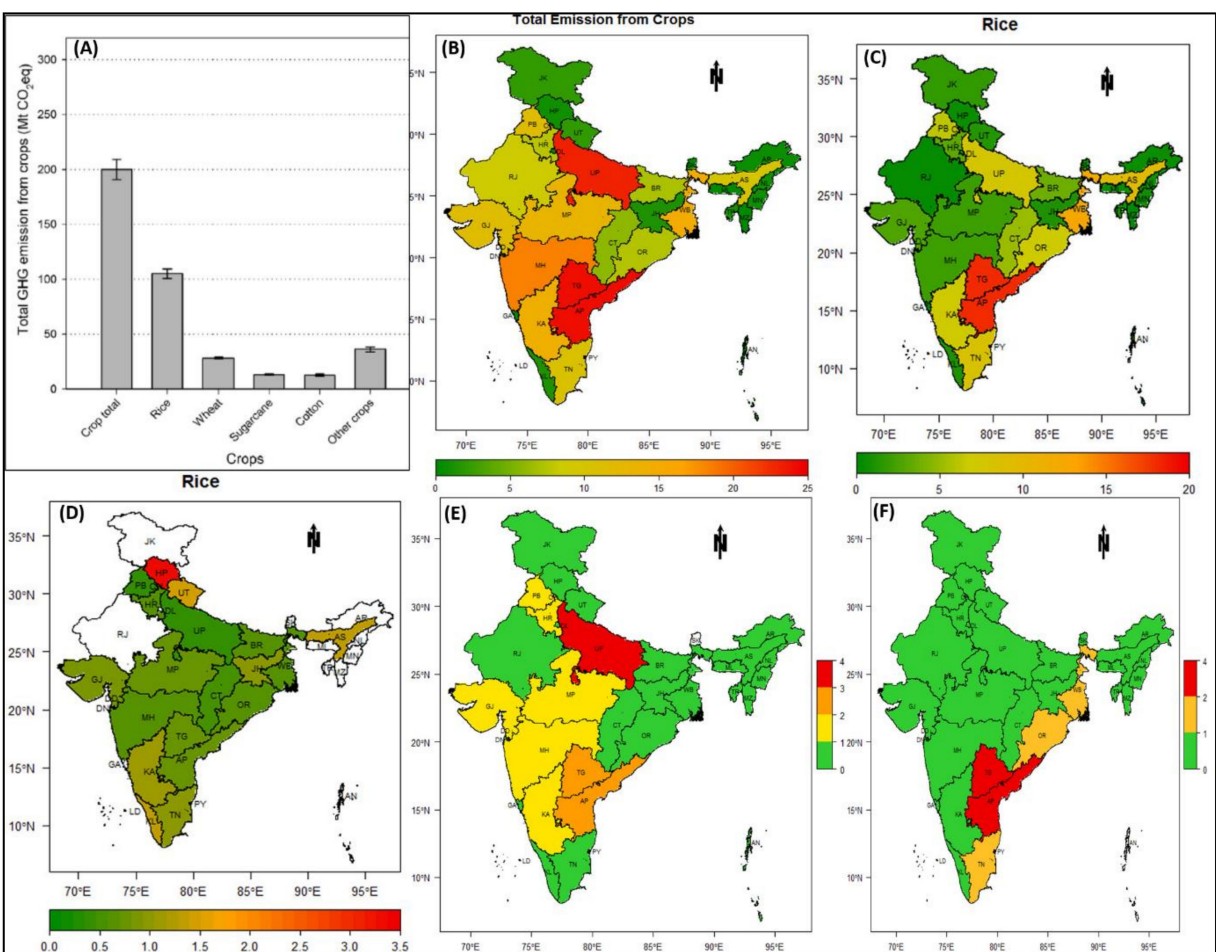

**Figure 2.** (**A**) Total national GHG emissions from crops. (**B**) State-wise distribution of total GHG emissions (Mt CO$_2$e) from crops and (**C**) from rice specifically. (**D**) GHG Emission intensity (kg CO$_2$e/kg product) of rice. (**E**) Spatial distribution of GHG mitigation potential (MtCO$_2$e per year) through improved fertilizer management and (**F**) through improved water management in rice. Source: Sapkota et al. [46].

In a study conducted in Nepal under the Asian Development bank funded project, DSR technology showed an impact in terms of the reduction in greenhouse gas (GHG) emissions by at least 8% considering only the tractor hours. Differential response of DSR-adapted rice varieties towards GHG emission was observed in Cambodia; the lowest methane emission was noted with CAR14 (66 to 68 kg/ha), compared to Phka Rumduol (80–98 kg/ha) (Figure 3A). Furthermore, the short duration rice variety CAR14 was considered a climate smart rice variety contributing very less to global warming (Figure 3B). Similarly, in Indonesia 46% [47] and in China 54% [48] lower global warming potential was reported in DSR compared to PTR.

A study conducted by Pathak et al. [44] showed that if the total cultivated area in Punjab is converted to DSR, the global warming potential will be reduced by 33%, and if half is converted to DSR, the global warming potential will be diminished by 16.6% of the current emission. Studies conducted in china showed that the global warming potential was about 76.2% lower for dry DSR, whereas it was 60.4% lower for wet DSR than for the PTR system of rice cultivation [49]. A reduction of around 43% cumulative methane emissions was observed from the no-tilling cultivation experiment compared to the PTR, indicating a more oxidative nature of plow layer in no-tilling cultivation [50]. Estimates suggested that the global warming potential for the rice-based cropping system can be reduced by a quarter if we replace conventional PTR by DSR in the Indo-Gangetic Plains. Moreover, the non-puddled transplanting of rice reported to save 35% of the net life cycle GHG emission compared with the conventional PTR practice. In addition to the reduction

in the GHG emission from soil, this practice led to the increase in the soil organic carbon (SOC) content [51]. Furthermore, the SRI (system of rice intensification) cultivation practice also reported low net GHG emission, and 1.5 times greater $N_2O$ emission due to the increase in the soil aeration. According to the IPCC, 2013 [52] report on climate change, the wetland rice production system contributes nearly 12% of the anthropogenic methane and 55% of the agriculturally-sourced GHG emissions in the world. Liu et al. [53] recorded about 54% more methane emissions from conventional PTR fields than the DSR rice, whereas the $N_2O$ emissions under PTR were reduced by about 49%. Similarly, Chakraborty et al. [28] conducted a meta-analysis on worldwide available data and reported that the methane emissions was higher under conventional PTR compared to the novel crop establishment practices. The increasing methanogenesis under puddled transplanted system of rice cultivation may be due to the reduced soil percolation leading to the reduction in the flow of oxygen-containing water [54], hence, resulting in the low emissions of $CH_4$ to the atmosphere.

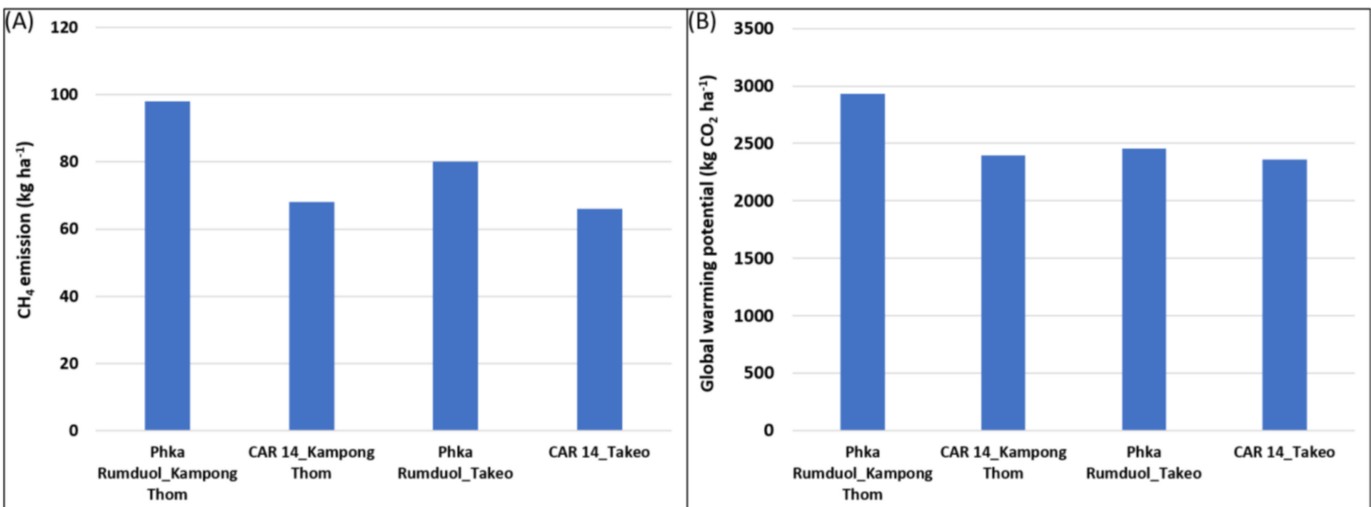

**Figure 3.** (**A**) Comparison of methane emissions and (**B**) global warming potential (GWP) using the two most cultivated varieties in Kampong Thom and Takeo, Cambodia.

### 2.3. Mechanization

Farm mechanization plays a significant role in increasing land, water and labor efficiency in agriculture. Efficient irrigation system machines, direct seeding/transplanting machines, powered sprayers, combine harvesters, dryers using biomass fuel, storage handling and high quality, advanced automated rice mill machines will be a basic need of Asian farmers in the near future. Mechanized DSR using the tractor-driven seed drills has enabled the seed sowing at an optimum soil depth of 2–3 cm in addition to a reduction in the seed rate from 80–200 kg ha$^{-1}$ to 20–25 kg ha$^{-1}$, which resulted in overcoming the lodging problems and the spikelet sterility [8]. The mechanization saves time and labor, reduces production cost and postharvest losses, increases grain quality and yield and generates employment for youths. Furthermore, there are certain important constrains to mechanization in agriculture, which are included in the following sections.

#### 2.3.1. Small Farm Size

Small scale farming is a big constrain in agricultural mechanization as it is against the principle of "economies of scale." Small scale local farming system could be inappropriate for import-based technology transfer strategy as most of the machines were developed in countries with large farm holdings.

### 2.3.2. Machinery

There is a lack of durable, light weight, compact, low-power, eco-friendly, multi-purpose and marketable machines that could meet farmers' operational needs at an affordable price. The prices of acquiring, maintaining and repairing these machines, using imported spare parts instead of locally-available materials in fabricating machines and ensuring the availability of spare parts, continue to stay at levels unaffordable to most farmers. The other factors include access to finance and market, seasonal demand of machine, limited supply chain, limited access and knowledge, prohibitive trucking and shipping rates, competition from imported products, irrational taxes, duties for raw materials and fabrication machines and improper coordination between demand and supplies.

### 2.3.3. Extension Workers

Extension workers play an important role in technology transfer. The lack of extension staff for a big number of client-farmers, lack of adequately trained personnel with the required technical expertise, communication skills and trainings and low capability to integrate the technology into mechanized farming system would likely end up in the non-adoption of mechanized technologies.

### 2.3.4. Inadequate Support Services

The lack of support services to ensure machine's acceptability to farmers, limited access to credit, ineffective marketing systems, limited cooperatives and associations and lack of entrepreneurs and their coordination with farmers have been continuing constraints in promoting agricultural mechanization.

### 2.3.5. Policy Constraints

Import restrictions, government priorities, mainstream in national development programs, investment in technology and logistics, unstructured tariff and taxation systems have had negative effects on adopting and promoting mechanized agriculture systems.

### 2.3.6. Knowledge

Knowledge and awareness of the rising need for farming mechanization by smallholder farmers is not sufficient. Further constrains include the mindset of farmers, resistance to change old practices, lack of curiosity, education, proper understanding and a lack of infrastructure and industry.

### *2.4. Early Planting of Second Crop*

DSR seems to be more suitable in the multiple cropping program due to early crop maturity by 7 to 10 days [55]. This early crop maturity allows timely planting of a second crop. Some farmers can even raise a third rice crop with supplemental irrigation in the Long An Province, Vietnam during December to February [56]. Furthermore, DSR shortens the cropping cycle as direct sowing prevents the transplanting shock to the seedlings [57].

### *2.5. Cost of Cultivation*

Dry-DSR has been reported to reduce the cultivation cost ranged from 6% to 32% (i.e., about US$29–125 ha$^{-1}$) and wet DSR from 2% to 16% (i.e., about US$8–34 ha$^{-1}$) [8]. Tripathi et al. [16] reported a lower cost of cultivation in the Haryana state of India largely due to the lower expenses on labor wages (6.62%), machine use (41.34%) and irrigation (22.45%). The benefit–cost ratio was higher (2.92) in DSR compared to the PTR system (2.61). Similarly, Pandey et al. [43] revealed that the profitability under DSR was higher than the PTR system of rice cultivation due to significant reduction in the cost of tillage operations. The cost required to produce one kilogram of rice was 5.68 and 6.34 rupees in DSR and PTR, respectively. The grain production cost was 10.44% lower in DSR compared to the PTR system. The cost-benefit analysis presented a higher ratio for the mechanized DSR than the traditional PTR system, with a benefit–cost ratio of 1.40 and yield of 4.2 t ha$^{-1}$ under DSR

system and a benefit–cost ratio of only 1.08 under the PTR system of farming used at the project sites in Nepal, with the same yield (Table 4). Based on the calculations, the grain yield potential of up to 6.0 t ha$^{-1}$ with a net profit up to NPR 62,000 and benefit–cost ratio of 2.0 is achievable under better crop management practices combined with the mechanized DSR technology. The experiments conducted in Nepal under mechanized DSR involving machine-operated boom sprayer and combined harvesting reported a 25% reduction in the total cost compared to the PTR system of rice cultivation. Seeds used decreased from 80 kg ha$^{-1}$ under PTR to only 45 kg ha$^{-1}$ under mechanized DSR; thus, again reducing the production costs and contributing to higher income. The use of a seed drill led to labor reduction and saving in the fuel costs, ultimately reducing the production costs and raising the income. Dhakal et al. [58], in an on-farm study, reported that the benefit–cost ratio under DSR was 2.0, which was higher than 1.63 for PTR and was considered a better alternative. Similarly, Bairwa et al. [59], in an on-farm assessment of DSR technology in the humid south-eastern plain of Rajasthan (India), reported 12% and 28% high net return (Rs 51,968 ha$^{-1}$) using PTR, respectively, and a 2.25 benefit–cost ratio using DSR.

**Table 4.** Comparison of the cost–benefit analysis of the area using DSR and the area using the traditional puddled system of rice cultivation, Nepal.

| Activities | Conventional Practice | | | | Mechanized DSR | | | |
|---|---|---|---|---|---|---|---|---|
| | Quantity | Rate (NRs.) | Amount (NRs.) | Amount (US$) | Quantity | Rate (NRs.) | Amount (NRs.) | Amount (US$) |
| Seed (kg) | 45.00 | 45 | 2025 | 17.00 | 45.0 | 45 | 2025 | 17.00 |
| Nursery raising | - | - | 3800 | 32.00 | - | - | - | - |
| Field preparation | - | - | 10,200 | 86.00 | - | - | 5100 | 43.00 |
| Transplantation/sowing in DSR | - | - | 16,000 | 135.00 | - | - | 3000 | 25.00 |
| Fertilizer management | - | - | 10,220 | 86.00 | - | - | 13,240 | 111.00 |
| Water management | - | - | 10,125 | 86.00 | - | - | 12,750 | 107.00 |
| Puddling Irrigation (h) | 20.00 | 125 | 2500 | 21.00 | - | - | - | - |
| Irrigation pre-monsoon, 2 times (h) | - | - | | - | 50.0 | 125 | 6250 | 53.00 |
| Irrigation during monsoon, 3 times (h) | 45.00 | 125 | 5625 | 47.00 | 36.0 | 125 | 4500 | 38.00 |
| Labour for pipe laying and irrigation | 5.00 | 400 | 2000 | 17.00 | 5.0 | 400 | 2000 | 17.00 |
| Plant protection (weed, insect, pest management) | - | - | 9400 | 79.00 | - | - | 11,786 | 99.00 |
| Harvesting and threshing | - | - | 18,525 | 156.00 | - | - | 12,800 | 108.00 |
| Depreciation of tools, equipment | - | - | 1500 | 13.00 | - | - | 2000 | 17.00 |
| Total Cost of rice/paddy production (NRs.) | - | - | 81,795 | 688.00 | - | - | 62,701 | 527.00 |
| Grain Yield(kg) | 4200 | 20 | 84,000 | 706.00 | 4200.0 | 20 | 84,000 | 706.00 |
| Straw yield (trolley) | 2 | 2000 | 4000 | 34.00 | 2.0 | 2000 | 4000 | 34.00 |
| Total income (NRs.) | - | - | 88,000 | 740.00 | - | - | - | - |
| Net Profit, (NRs.) | - | - | 6205 | 52.00 | - | 2000 | 25,299 | 213.00 |
| Benefit/Cost ratio | - | - | 1.08 | 1.08 | - | - | - | - |
| Labour | 101.0 | 400.0 | 40,400.0 | 340.00 | 17.0 | 400.0 | 6800.0 | 57.00 |
| Tractor hour (h) | 10.75 | - | 17,425.00 | 147.00 | 9.75 | - | 23,100 | 194.00 |
| Irrigation hour (h) | 90 | - | 10,625 | 89.00 | 86.0 | - | 10,750 | 90.00 |

NRs: Nepalese Rupee.

## 2.6. Grain Yield and DSR-Adapted Rice Varieties

Despite all the benefits of DSR technology, the grain yields showed variability in some of the growing regions, especially due to the uneven and poor crop stand, more weeds, more spikelet sterility, lodging problem and poor knowledge of water-nutrient-weed management. Furthermore, the rice varieties cultivated under DSR are mainly bred and selected for PTR system of rice cultivation. The yield under DSR cultivation conditions depends on the effective and precise use of nematicides and herbicides as well as proper and timely supply of water and nutrients. The grain yields under DSR involving DSR-

adapted rice varieties reported to be varied from 4.5 to 6.5 t ha$^{-1}$, which is approximately 2 to 3 times higher than the yield obtained with the traditional cultivated upland rainfed rice varieties, but 20–30% lower than the yield obtained with the lowland varieties grown under PTR conditions [60,61]. For the sustainability of DSR system of rice cultivation, targeted breeding efforts must be adopted for the long-term sustainable and improved yields through the development of rice varieties that do not show any yield decline under DSR cultivation system.

Guimaraes [62] reported DSR-adapted rice varieties with yield potential of 6.0 t ha$^{-1}$ in Brazil. The other successful example of DSR-adapted rice varieties include MAS 26, ARB 6 and MAS 946-1 from the UAS (University of Agricultural Sciences), Bangalore [63]; Han Dao (HD277, HD297 and HD502) from CAU (China Agricultural University), China [53]; CR Dhan 200, CR Dhan 201, CR Dhan 202, CR Dhan 203, CR Dhan 205 and CR Dhan 206 from NRRI (Cuttack); Sahod Ulan 12, APO and CT-6510- 24-1-2 from IRRI (Philippines) [64]; Magat (IR64616H) [65]; CAR 14 from Cambodia; and Tarahara 1 from Nepal. Most of these DSR-adapted rice varieties reported to possess semi-dwarf height, earliness, early vigor, improved yields with better grain quality, resistance to biotic stress (blast, bacterial blight), drought tolerance and an ideal plant type with increased lodging resistance and erect upper leaves, indicating their suitability for adoption under DSR [53,66].

## 3. Risks Associated with DSR

In addition, there are some risks associated with the shifting of PTR system to DSR, which include (1) high weed infestation, (2) increases in the soil-borne pathogens such as nematodes, (3) issues related to nutrient uptake.

### 3.1. Weed Competitiveness

Most often, notwithstanding the high yield potential of the varieties, actual yields were observed to be much lower than the expected one because of poor management of weeds, one of the major issues in DSR system of rice cultivation [67]. In a DSR system, unlike other rice ecosystems, the transitional soil-water regime of dry and wet spells alternately generates a congenial microenvironment that prompts the emergence and faster growth of extremely competitive complex weed flora in many flushes. In these conditions, direct-sown rice seeds, when emerging, have no "head-start" over the weeds as transplanted rice seedlings usually do with a standing-water layer [68]. A mixed flora of weeds comprising mostly grassy weeds at the beginning of season and sedges and broad-leafed weeds in later stages of crop growth usually appears in DSR. The predominant grassy weeds are *Echinochloa colona*, *E. crus-galli*, *Digitaria sanguinalis*, *Eleusine indica* and *Dactyloctenium aegyptium*; sedges are *Cyperus iria*, *Cyperus rotundus*, *Cyperus difformis*, *Fimbristylis miliaceae*, *Alternanthera sessilis* and *Amaranthus spinosus*; and broadleaf weeds are *Eclipta alba*, *E. prostrata*, *Commelina benghalensis*, *Portulaca oleracea*, *Euphorbia hirta* and *Ludwigia parviflora*. A yield reduction of 48%, 53% and 74% under PTR, direct-seeded flooded and DSR cultivation system, respectively, was reported by Ramzan [69]. The season-long weed in DSR system may cause 80% grain yield reduction [70]. The other studies reported 58% and 82% yield reduction at a density of 40 and 215 weeds m$^{-2}$ [71,72], respectively. Therefore, up-scaling DSR cultivation requires comprehensive measures considering the compatibility of both the weed competitiveness/weed-suppressive ability of rice genotypes and efficacy of the manual and chemical weed control methods.

### 3.2. Nutrient Uptake

The lower yield under DSR may also result from decreased nutrient uptake by rice roots under DSR conditions as compared with PTR conditions. This situation requires the identification of the root structure and distribution to enable rice to have better nutrient uptake under DSR conditions. The root distribution in deep soil layer may help to improve the rice grain yield and NUE (nitrogen use efficiency) under DSR. It may improve photosynthesis and delay leaf senescence [73]. The increase in the number of nodal roots may

improve root nutrition absorption. Furthermore, this may play a major role in providing lodging resistance. Micronutrients, such as Fe and Zn, are even required in small amounts but they affect the photosynthesis, grain yield and the grain nutrient enrichment [74,75], in addition to their pivotal role in the vital processes such as protein synthesis, respiration and reproduction phase. The nutrient requirement in DSR is higher as compared to PTR because of the no-standing water, higher plant density and more biomass production in the vegetative phase [76].

### 3.3. Nematode Infestation

The cultivation of rice under DSR is likely to promote an increase in the root-knot nematode (*Meloidogyne graminicola*) in the rice root zone, thereby decreasing rice yield. *Meloidogyne graminicola* is causing severe damage under rain-fed lowland, upland [77] and irrigated rice cultivation conditions [78]. The nematode has been reported to cause 16% to 32% rice crop yield losses in the upland and rainfed rice [79] in India. Approximately 2.6% grain reduction was reported under upland conditions for every 1000 nematodes that are present around the young seedlings.

### 4. Recommended Package of Practices under DSR

The detailed information on the required cultivation practices (Figure 4), varieties, benefits of DSR cultivation practices and yield potential in diverse ecosystems under direct seeded rice cultivation system are provided in the Table 5.

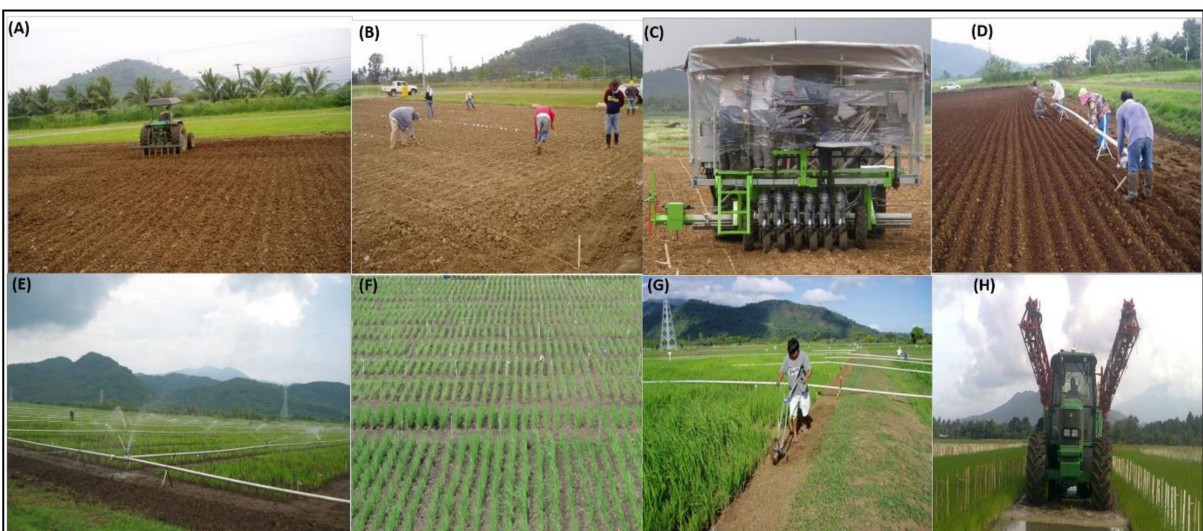

**Figure 4.** Agricultural practices followed in direct seeded rice cultivation systems: (**A**) land preparation, (**B**) manual seed sowing, (**C**) seed showing using mechanized seed drill, (**D**) installation of sprinkler irrigation system, (**E**) irrigation through sprinkler system at seedling stage, (**F**) field view of DSR field at seedling stage, (**G**) manual weed control using wheel hoe, (**H**) mechanized weed control using boom tractor sprayer.

**Table 5.** Detailed information on varieties, required cultivation practices and their benefits and yield potential in diverse ecosystems under direct seeded rice cultivation system.

| Country | Ecosystem | New Varieties | Water Savings | Yield | Labor | Weed Management | Risk and Return |
|---|---|---|---|---|---|---|---|
| India | Irrigated and rainfed | CR dhan200, CR dhan201, CR dhan202, CR dhan203, CR dhan205, Anagha | Highest yield per unit of water used among all water-saving rice technologies | 3.5–6.1 t ha$^{-1}$ | Labor-saving input is the key driver in adopting the technology | Proper land preparation, clean seed, pre/post emergence herbicides, cultural operation | Small sowing window, high rainfall just after sowing reduces germination, establishment, higher return in case of scattered initial rain |
| | Rainfed: Water limited uplands and shallow low lands in Eastern region (Jharkhand and Odisha) | Sahbhagi Dhan | Limited opportunity for water savings | 2.0–4.5 t ha$^{-1}$ depending upon ecosystem and water availability; 1.0 t ha$^{-1}$ higher yield over currently grown varieties under water shortage | Labor saving, better opportunities for mechanized sowing and the crop management | Proper land preparation, clean seed, pre/post emergence herbicides, cultural operation | Reduced risk of crop establishment if monsoon is delayed; Possible early harvest allows a sequence crop with residual moisture |
| | Rainfed: Drought prone eastern region (Odisha, Karnataka) | Apo, Sahbhagi dhan, CR dhan203, CR dhan205 and Sharada | Limited opportunity for water savings | 2.0–4.5 t ha$^{-1}$ depending upon ecosystem and water availability; 1.0 t ha$^{-1}$ higher yield over currently grown varieties under water shortage | Labor saving, better opportunities for mechanized sowing and the crop management | Proper land preparation, clean seed, pre/post emergence herbicides, cultural operation | Small sowing window, high rainfall just after sowing reduces germination, establishment, higher return in case of scattered initial rain |
| | Irrigated, Odisha | MTU1010, DRR dhan 42, DRR dhan 44 | 25% irrigation water savings, water use efficiency (WUE) of 3.84 kg ha$^{-1}$ mm$^{-1}$ (3.37 kg grain ha$^{-1}$ mm$^{-1}$ for conventional system) | Aerobic: 2.4–4.2 t ha$^{-1}$ (25–30% lower than conventional) | Labor-saving opportunities with mechanized DSR planting | Preemergence herbicide followed by one manual weeding at 3 WAS increase yield; Ground nut or mung bean sequentially can enhance yield (4.3 to 4.6 t ha$^{-1}$) | Small sowing window for direct seeded rice |

**Table 5.** *Cont*.

| Country | Ecosystem | New Varieties | Water Savings | Yield | Labor | Weed Management | Risk and Return |
|---|---|---|---|---|---|---|---|
| Nepal | Shallow rainfed and hilly upland lowland irrigated and rainfed area | Sukha dhan 1, Sukha dhan 2, Sukha dhan 3 | Limited opportunity for water savings | 3.0–4.3 t ha$^{-1}$, 1.0 t ha$^{-1}$ more yield over cultivated popular varieties under water shortage | Labor-saving opportunities in rainfed and irrigated areas with mechanized seeding | Weed management using butachlor at 1.5 kg a.a. ha$^{-1}$ sprayed 2 WAS and hand weeding 4 WAS most effective | Short window for direct seeding, higher return |
| Pakistan | Punjab and Sindh | IR79597-56-1-2-1, IR80416-B-32-3 | At least 25% savings | 5.0–5.8 t ha$^{-1}$, an increase by 14% over currently popular varieties | Labor-saving opportunities with mechanized DSR planting | Weed management successfully addressed with herbicides (ethoxysufuron and sodium 2, 6 bis-benzoate) and increased yield | Incremental return of $402 ha$^{-1}$ |
| Philippines | Tarlac, Nueva Ecija, Bulacan | Apo, Sahod Utan 1, Sahod Ulan 12 | - | Apo: 4.0–5.5 t ha$^{-1}$, 2.0 t ha$^{-1}$ in Bulacan Sahod Ulan 1: 5.26 t ha$^{-1}$ in Bulacan | Labor-saving opportunities with mechanized DSR planting | - | Return as good as irrigated lowland rice. An effective alternative for rainfed and water short areas |

h: hectare, t = ton, WAS = week after seeding; Source: RETA 6276: Developing and disseminating water-saving rice technologies in South Asia, 17 technical papers, http://www.adb.org/projects/documents/devdissemination-climate-resilient-rice-varieties-for-water-short-areas-of-sa-sea-17-papers-tacr accessed on 22 December 2020, Modified from ADB brief, Developing and disseminating water-saving rice technologies in Asia (2016). ISBN 978-92-9257-523-6.

*4.1. Crop Establishment*

- Time of sowing: dibble-seeding during the first to fourth week of January in the dry season and from the second week of June to first week of July in the wet season, depending on rainfall pattern.
- Seed rate: 25 to 50 kg ha$^{-1}$ in mechanical (pneumatic seeder) or manual seeding.
- Spacing: 20 cm $\times$ 20 cm (20 cm between lines, 20 cm between hills) in both the wet and dry seasons.

*4.2. Water Management*

Physical symptom: apply irrigation at tip rolling of young leaf in the morning or evening, or upon development of hairy cracks in the surface soil.

Piezometric device: 0–25 kPa soil moisture content at the root-zone depth (depending on variable soil physical structure).

*4.3. Weed Management*

Various studies were conducted in different collaborative countries, such as Bangladesh, India, Nepal and the Philippines, and the recommended package of practices are as follows:

- Select a variety with faster initial growth, high early vegetative vigor and early canopy cover ability.
- Use clean paddy seed for the sowing and machines for the land preparation.
- Maintain clean bunds and irrigation canals.
- Practice off-season tillage/summer plowing.
- Practice stale seedbed sowing (depending on the quantity of initial water availability).
- Use preemergence weedicides: pendimethalin at 2.5–3.0 L ha$^{-1}$, pyrazosulfuron at 200 mL ha$^{-1}$, pretilachlor at 1.0 L ha$^{-1}$ and butachlor at 1.5 kg a.i. ha$^{-1}$ 3–5 days after sowing.
- Use postemergence weedicides: bispyribac sodium (Nominee) at 250 mL ha$^{-1}$ at 15–25 DAG (days after germination) and azimsulfuron (Segment) at 35–50 mL ha$^{-1}$ at 15–20 DAG.
- For mechanized cultivation, one preemergence spray of pendimethalin at 2.5–3.0 L ha$^{-1}$/pyrazosulfuron at 200 mL ha$^{-1}$ and pretilachlor at 1.0 L ha$^{-1}$/butachlor at 1.5 kg a.i. ha$^{-1}$ at 3–5 DAS followed by one postemergence spray of bispyribac sodium (Nominee) at 250 mL ha$^{-1}$ at 15–25 DAG/azimsulfuron (Segment) at 35–50 mL ha$^{-1}$ at 15–20 DAG will lead to effective weed control.

In case of failure of application of pre-emergence weedicide because of too much rain after seeding, use at least one hand weeding or mechanical weeding (by rotary weeder/cono-weeder/hand wrecker at 12–15 DAG with post-emergence or at 20–25 DAG with pre-emergence) in addition to the application of an effective pre-/postemergence weedicide.

*4.4. Herbicide Resistance in Weeds: Alleviating Strategy*

(1)   Herbicides rotation with different modes of action,
(2)   Use of the herbicide mixtures and recommended dose and rates of herbicides,
(3)   Pulling out weeds which have escaped herbicide application,
(4)   Preventing the spread of resistant weeds from one area to another through farm implements/machinery,
(5)   Adoption of IWM (integrated weed management) practices,
(6)   Adoption of crop diversification and crop rotations.

## 5. Conventional Breeding Efforts

Conventional breeding has been reported to be very successful in improving the biotic and abiotic stress resistance, productivity and grain quality in many cereal crops. Biparental and multiparent crosses are extensively used in varietal improvement programs

to combine the desirable characteristics of the ideal parents. The objectives of the crop breeding improvement programs have become more and more diverse, and it is not possible to achieve the target gain through biparental mating. To overcome the genetic limitation of restrictive recombination events in biparental crosses, the use of 16 to 32 parents in the multiple crosses could be efficiently used in rice crop breeding improvement program [80]. Theoretically, the multi-parental crosses lead to the recombination of genes from many parental strains involving intermating of the $F_1$s in succeeding generations. The practical limitation of using multi-parents in a crossing program may include the incorporation of a number of undesirable alleles, which may likely disturb the ideal genetic background of the varieties that took several years or even decades to assemble.

Sandhu et al. [81] evaluated the performance of conventionally bred DSR-adapted rice genotypes in a series of 23 experiments conducted across different location in Bangladesh, India, Lao-PDR Nepal and the Philippines between 2014–2017. High yielding promising breeding lines with improved grain yield and adaptability under DSR cultivation conditions were identified. The mean grain yield of the selected genotypes was 5.0 t ha$^{-1}$ across seasons, years and different locations under DSR conditions, indicating the DSR adaptability and high grain yield potential of the promising genotypes across diverse ecosystem. The better grain yield advantage of the promising genotypes over the currently existing and locally adapted rice varieties, indicates the suitability of selected genotypes to be released as variety for cultivation under DSR conditions (Table 6). Another study was conducted at IRRI, the Philippines with an objective to develop high-yielding and direct seeded adapted rice varieties utilizing biparental to multiparent crosses involving as many as six different parents [82]. A large number of crosses, involving biparent, triparent, quadraparent, pentaparent and hexaparent crosses, were attempted and evaluated from $F_2$ generation to the advanced yield trials involving very stringent phenotypic selection, including plant type, plant height, maturity, number of tillers, visual yield and grain type. The involvement of the higher number of parents in multi-parent populations resulted in higher grain yield advantages compared to the classical biparental population. The developed multi-parent breeding lines with improved grain yield, superior grain quality and better adaptability to DSR may have the potential to be released as varieties in different countries of South and South East Asia.

**Table 6.** Mean grain yield (Kg ha$^{-1}$) of the selected promising genotypes across diverse ecosystems under DSR conditions.

| Genotype | Philippines | Bangladesh | | Nepal | | India | | Lao PDR |
|---|---|---|---|---|---|---|---|---|
| | IRRI, Los Baños | BRRI, Gazipur | BRRI, Rajshai | NRRP, Hardinath | RARS, Tarahara | NRRI, Cuttack | IRRI SAH, Hyderabad | NAFRI, Laos |
| IR 91326-19-2-1-2 | 5204 (11%) [†] | 3661 (14.4%) [†] | - | 4830 (11.3%) [†] | 3430 (3%) [†] | - | 3433 (–) | - |
| IR 97041-8-1-1-1 | 5670 (20.1) [†] | 3290 (2.9%) [†] | 2991 | 4750 (9.4%) [†] | 4332 (30.1%) [†] | - | 4580 (18.8%) [†] | - |
| IR 92521-146-3-3-2 | 5328 (13.6%) [†] | 3550 (10.9%) [†] | - | 4402 (1.4%) [†] | 3800 (14.1%) [†] | 2815 | 3891 (1%) [†] | 3500 (7.0%) [†] |
| IR 93835-70-2-2-1 | 4830 (2.9%) [†] | 2080 (–) | - | 4605 (6.1%) [†] | 3380 (1.5%) [†] | - | 4798 (24.5%) [†] | 2889 (–) |
| Vandana * | 4690 | 3195 | - | 3750 | 2205 | 1420 | 2100 | - |
| UPLRi 7 * | 4680 | 3485 | - | 3850 | 3220 | 1508 | 1802 | - |
| IRRI 132 * | 4110 | 3305 | - | 3822 | 3010 | 1550 | 2052 | - |
| BRRIdhan 29 * | - | 3200 | - | - | - | - | - | - |
| Shabhagi dhan * | - | - | - | - | - | 2735 | 3855 | - |
| Hardinath * | - | - | - | 4340 | - | - | - | - |
| Tarahara * | - | - | - | - | 3330 | - | - | - |
| TDK1 * | - | - | - | - | - | - | - | 3270 |
| Trial Mean | 4992 | 3796 | 2870 | 3780 | 3127 | 3927 | 4423 | 3420 |
| LSD (0.05%) | 954 | 396 | 242 | 883 | 386 | 482 | 729 | 721 |

\* Check variety, [†] percentage (%) grain yield advantage over the respective check in each location, – indicates no yield advantage over the respective check in each location; Vandana (Philippines), BRRIdhan 29 (Bangladesh), Hardinath (Hardinath, Nepal), Tarahara (Tarahara, Nepal), Shabhagi dhan (India), TDK1 (Lao PDR). Source: Sandhu et al. [81].

To date, many high yielding rice varieties from the conventional [83] and genomics-assisted breeding program [84] have been released. Even though the conventional breeding approach has been proven to be an effective approach for the development of novel genetic variants [85], this approach also suffers from the problem of the long time required to develop homozygous breeding lines, linkage drag and the low efficiency, which impede the success of conventional breeding [86].

## 6. Use of Modern Breeding Tools to Achieve Higher Crop Productivity

### 6.1. Genomics-Assisted Breeding Efforts

Development of DSR-adapted multi-stress tolerant rice varieties is required to better sustain the water-labor shortage and grain yield losses from unpredicted climate-related events. The schematic representation of the steps involved in the development and release of direct seeded adapted rice varieties is presented in Figure 5. Over the last 10 years, research at IRRI, various new donors, traits and QTL have been identified for traits that increases rice adaptation as well as grain yield under DSR conditions. At IRRI, QTL for early uniform emergence ($qEVV_{9.1}$, $qEMM_{1.1}$, $qEMM_{11.1}$), higher root length density ($qRHD_{1.1}$, $qRHD_{5.1}$, $qRHD_{8.1}$) facilitating higher N, P, K uptake ($qN_{5.1}$) in variable anaerobic-aerobic soil conditions, lodging resistance ($qLDG_{4.1}$) and grain yield under direct seeded situation ($qGY_{1.1}$, $qGY_{6.1}$, $qGY_{10.1}$) [87,88], were identified and have been used in the breeding program for the development of better aerobic rice varieties following marker-assisted selection. In addition to the above-mentioned QTL, QTL for increased yield under reproductive stage drought stress ($qDTY_{1.1}$, $qDTY_{2.1}$, $qDTY_{3.1}$, $qDTY_{12.1}$) has been used in the introgression program [82]. These developed marker-assisted-derived breeding lines are available for testing as well as identified QTL, and markers are available for rice researchers to undertake further introgression programs. The comparative performance of DSR lines carrying the multiple QTL/genes for DSR wider adaptability traits and biotic/abiotic stresses over the popular checks has been represented in Table 7. In addition, at IRRI, new donors for some of the important traits improving yield and adaptability under DSR have been identified. These donors include Ashmber, R146, NCS237, N22, Shangyipa, Solomon and WP65 for better germination from a soil depth below 4 cm, WAB880-1-27-9-2-PI-HB for high nutrient uptake, Dular for better root length density, WAB880-1-27-9-2-PI-HB for more lateral roots, Kali Aus and Kalinga 3 for higher percent lateral roots, Basmati 370 for tolerance to Fe deficiency, Facagro 64, A 69-1, Baduie, Jagli boro for tolerance to Zn deficiency and CG14, IR72 for nematode tolerance.

Recent developments in the identification of major QTL/genes and their successful introgression in different elite genetic backgrounds to develop improved varieties tolerant to different individual stresses indicated that, with the advent of new marker technology, the development of multi-stress-tolerant varieties is feasible. Such varieties, once developed, can help farmers to overcome yield losses and better farm income under changing climatic conditions. Sandhu et al. [89] identified QTL for the grain yield, yield and root-related traits, and various agronomic traits under direct seeded aerobic conditions using two $F_{2:3}$ mapping populations (HKR47 × MAS26 and MASARB25 × Pusa Basmati 1460). The significant positive correlation and co-location of genomic regions associated with particular traits such as root traits and grain yield indicating the role of these root traits in improving the grain yield under direct seeded aerobic cultivation conditions possibly through efficient water and nutrient uptake. The relationship between phenotypic root plasticity, nutrient uptake and grain yield stability across variable growing conditions and environments was studied in two $BC_2F_4$ mapping populations derived from the Aus 276 × MTU1010 and the Kali Aus × MTU1010. Hotspots were identified for multiple root plasticity traits in both the populations and colocation of genomic regions associated with grain yield and root plasticity traits was detected [90].

**Table 7.** Performance of multiple trait introgression lines over the checks under DSR conditions.

| Designation | PHT(cm) | QTLs/Genes | No. of QTLs/Genes | GYKGPHA_DSR | % Yield over MTU1010 | %Yield Increase over UPLRi7 |
|---|---|---|---|---|---|---|
| IR 129477-1629-210-4-4-4 | 103 | $qDTY_{2.1}+qDTY_{3.1}+xa5+Xa21+BPH3+Pita+qAG_{9.1}+qNR_{5.1}+qRHD_{1.1}+qEMM_{1.1}$ | 10 | 5338 | 70 | 64 |
| IR 129477-1232-81-2-1-3 | 108 | $qDTY_{3.1}+GM4+Pita+qAG_{9.1}+Pi9+qNR_{5.1}+qRHD_{1.1}+qEVV_{9.1}$ | 8 | 4630 | 47 | 37 |
| IR 129477-3873-297-3-3-2 | 105 | $qDTY_{1.1}+Xa4+xa13+BPH3+GM4+Pita+qAG_{9.1}+Pi9+qNR_{5.1}+qRHD_{1.1}$ | 10 | 4466 | 42 | 32 |
| IR 129477-1629-14-1-1-1 | 106 | $qDTY_{3.1}+Xa4+Xa5+Xa21+BPH3+Pita+qAG_{9.1}+Pi9+qNR_{5.1}+qRHD1.1+qEMM_{1.1}$ | 11 | 4308 | 37 | 27 |
| IR 129477-1629-14-1-2-2 | 105 | $qDTY_{3.1}+Xa4+xa5+Xa21+BPH3+Pita+qAG_{9.1}+Pi9+qNR_{5.1}+qRHD_{1.1}+qEMM_{1.1}$ | 11 | 4061 | 29 | 20 |
| IR 129477-4026-249-16-6-3 | 91 | $qDTY_{3.1}+qDTY_{12.1}+Xa4+Xa21+BPH3+GM4+qAG_{9.1}+qRHD_{5.1}+qEMM_{11.1}$ | 9 | 3702 | 17 | 10 |
| IR 129477-3343-500-36-3-4 | 101 | $qDTY_{3.1}+Xa4+xa5+xa13+GM4+Pita+qAG_{9.1}+qRHD_{1.1}+qEMM_{11.1}$ | 9 | 3635 | 15 | 7 |
| IR 129477-3425-307-5-6-3 | 96 | $qDTY_{1.1}+qDTY_{3.1}+Xa4+xa5+BPH3+Pita+qAG_{9.1}+Pi9+$ $qGY_{10.1}+qNR_{5.1}+qNR4.1+qRHD_{1.1}$ | 12 | 3502 | 11 | 4 |
| IR 129477-1813-191-5-7-6 | 94 | $qDTY_{1.1}+Xa4+BPH3+Pita+qAG_{9.1}+Pi9+qGY_{6.1}+qRHD_{1.1}+qEMM_{11.1}$ | 9 | 3475 | 10 | 3 |
| MTU 1010 | 100 | - | | 3143 | - | - |
| IR 09N538 | 105 | - | | 2524 | - | - |
| UPLRI 7 | 104 | - | | 3375 | - | - |

Note: PHT: Plant height (centimeter); GYKGPHA: Grain yield in kilogram/hectare. Source: Sandhu et al., 2021, Frontiers in Plant Science (accepted).

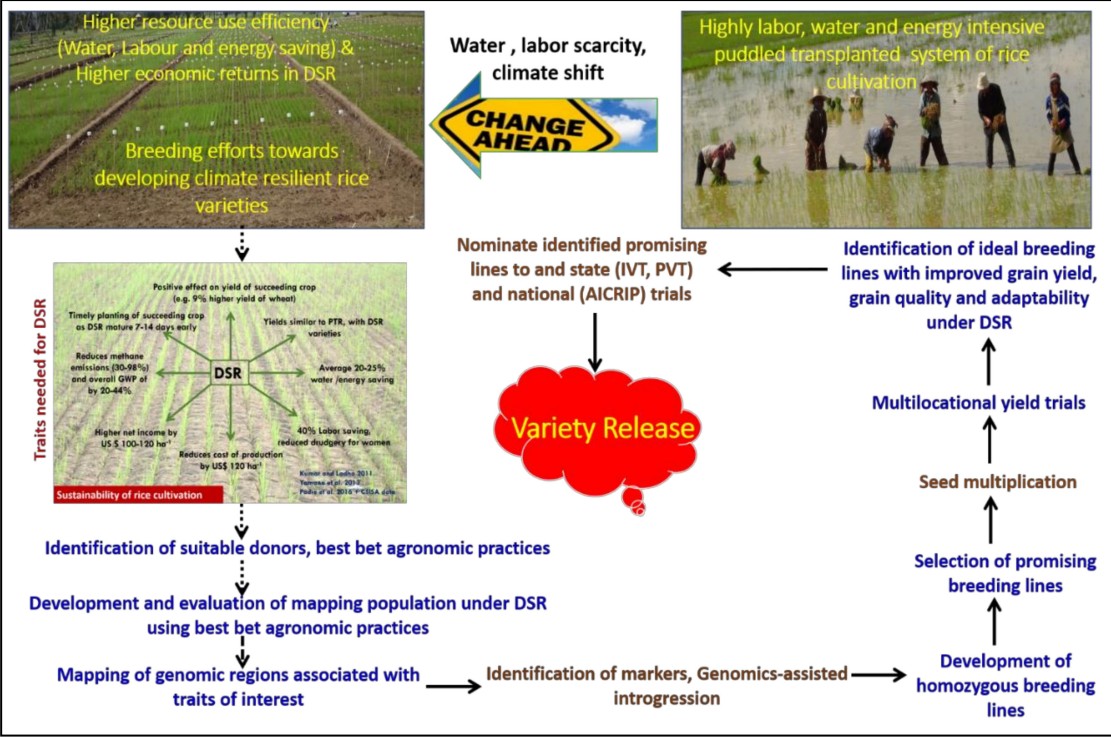

**Figure 5.** The schematic representation of the steps involved in the development and release of direct seeded adapted rice varieties. The identification of donors, traits, best bet management practices, development of suitable mapping populations and selection of promising breeding lines with improved grain yield, grain quality and adaptability are essential steps in the development and release of suitable DSR-adapted rice varieties. Identification of best management practices for the proper evaluation of breeding material underlies the success of DSR variety and technology.

To date, only very few studies on genome-wide association studies (GWAS) for root morphology, root traits improving nutrient uptake, grain yield and yield-contributing traits in rice under DSR cultivation systems have been reported. Recently, GWAS were conducted on a multi-parent complex mapping population derived from a genomics-assisted breeding program involving 5/6 parents [91,92]. The studies aim to identify the significant marker-trait associations (MTAs) for the traits, such as seedling-establishment traits, root traits improving nutrient-uptake, lodging resistance, plant morphology, yield and yield-related traits, providing yield improvement and grain yield stability under DSR. A total of 10 significant MTAs and 25 QTL associated with 25 traits [91] and 37 significant MTAs associated with 20 traits [92] were detected under direct seeded cultivation conditions. Significant and positive correlation across different populations and seasons was reported between grain yield, seedling-establishment traits, root and nutrient uptake-related traits and yield-attributing traits [91,92]. Utilizing the donors and the genomic regions identified in the previous studies conducted at IRRI, a genomics-assisted introgression program was initiated in 2012 at IRRI, with an aim to develop high-yielding, direct seeded adapted varieties [82]. The developed promising breeding lines with superior grain quality and better adaptability to DSR in addition to carrying 7–11 QTL/genes for various biotic and abiotic tolerance/resistance have the potential to be released as DSR varieties in different countries of South and South East Asia.

The advances including high throughput phenotyping, development of novel marker and new molecular breeding methods, development and use of multi-parent population have paved a new era of genomics-assisted breeding. These advances have led to the gradual shifting of the focus from phenotype-based selection in traditional plant breeding to the genotype-based selection [93] in genomics-assisted breeding. Sandhu et al. [82]

compared the grain yield performance of conventionally and genomics-assisted-derived breeding lines and observed that the genomics-assisted-derived breeding lines showed better results to improve yield, adaptability and multiple stress tolerance/resistance over traditional conventional breeding approach. In this way, the genomics-assisted breeding is very useful to get the desirable and necessary QTL/genes combinations without any unwanted genes thus minimizing the linkage drag around the target QTL/genes.

### 6.2. Genomic Selection for Higher Gain in Shorter Period of Time

Genomic selection (GS), an advanced form of marker-assisted selection (MAS), is currently considered as one of the most promising advanced tools for genetic improvement of the complex traits governed by multiple genes each with minor effects [94]. In genomic selection, the selection decision is based on the calculation of genomic estimated breeding values (GEBVs) by using the genome-wide markers implemented in a training population derived from a set of individuals which have genotyped and phenotyped both [94]. GS has a significant advantage over MAS in capturing both major and minor gene effects using genome-wide markers; additionally, no phenotyping is required during later breeding stages [95]. However, the successful implementation of GS models in various crops is still ongoing and debatable. Breeding efficiency is very high compared to traditional pedigree breeding method if successful GS has been implemented in the respective crop, as the selection is directly proportional to the GEBV accuracy.

GS has been successfully implemented in crops such as maize and wheat for grain yield and other quantitative traits in biparental and double haploid population [96,97]. Additionally, a few of the studies have reported GS implementation involving multiple parents by using genome-wide SNP and diversity array technology (DArT) markers in the crops, such as barley and wheat [98,99]. In rice, GS had been deployed for the agronomic and yield related traits such as grain yield, grain number, thousand-kernel weight [100,101], plant height, heading date, tiller number and panicle length [102,103]. However, genomic selection accuracy has been reported to be the highest (r > 0.9) for rice heading date, ranging from 0.25–0.90 in various studies [102,103]. There has been recent progress and development in plant growing techniques that can reduce the time to harvest and significantly accelerate breeding programs by reducing the generation time [104]. Speed breeding is an automated system where plants are grown under controlled conditions with a continuous source of light (22 h daylight) and optimal temperature. The advantage of speed breeding in the significant reduction of generations has been proven for crops such as wheat, barley, oat, chickpea, peanut and brassica species [104]. Combining GS with speed breeding may provide more intense, quicker and more efficient selection, which allows a higher genetic gain per year [105]; however, these novel breeding programs required adequate testing and standard operating procedures for its optimal utilization in various plant breeding programs.

Most of the GS studies reported single trait genomic prediction in various commercial breeding schemes in different crops, including rice; however, multiple trait genomic prediction should be extensively examined, keeping in view the development of the water saving dry direct seeded rice varieties in a shorter period of time, which have wider adaptability for aerobic ecology as well as having tolerance for abiotic and biotic stresses prevailing under a DSR system.

### 6.3. Mining of Novel and Superior Alleles through Haplotype Breeding

Under the current scenario of climate change, there is an imperative need to reinvent effective strategies to develop high yielding, climate-resilient rice varieties with superior grain quality traits. A clear understanding of the responsible genes and molecular mechanisms underlying stress tolerance is a prerequisite for enhancement of crop stress tolerance [106]. In this perspective, diverse germplasm collections rich in landraces, wild germplasm and breeding lines serve as a potential source of haplotype diversity for important genes responsible for abiotic stress tolerance [107]. Several key genes associated with

rice grain yield and related traits have been functionally characterized in the past. About 63 grain quality and 367 stress responsiveness genes have been cloned and functionally validated [108]. Rice literature revealed a haplotype diversity of about 120 key genes associated with grain yield and quality traits across the 3K rice genomes panel and identified superior haplotypes for genes influencing 10 target traits [109]. Haplotype-based genomic selection using NGS-based high density DNA arrays has the potentiality to estimate the predictive ability of different GS models applied in various crops [110]. However, the combined use of haplotypes and GS predominantly used and reported in self-pollinated crops such as wheat and soyabean, where higher LD values exists and which favors the identification of haplotype and haplotype blocks, consists of a greater number of alleles [111].

*6.4. Creation of Novel Genetic Variation through CRISPAR-CAS Technology*

The clustered regularly interspaced short palindromic repeats (CRISPR)/CRISPR-associated protein (Cas9) technology is a potent genome-editing method through which targeted modification of a plant genome can be done precisely and will be helpful for targeted genetic engineering for manipulating gene functions in plant. Through the tar gated genome editing approach, multiple traits can be modified precisely, which will be helpful in pyramiding of multiple genes. Using CRISPR technology, the negative regulators associated with disease resistance, abiotic stress tolerance and grain development can be knocked out in order to obtain higher yield, plant resistance against various pathogens and plant tolerance against, for example, drought and salinity [112,113]. Most of the agronomic traits, including grain yield, are complex in nature and controlled by polygenes/QTL. For example, in rice grain, the yield is functionally characterized, and the number of genes/QTL the trait has been reported in rice literature [114]. Introgression of single or multiple QTL can improve the grain yield; however, in some of the reports, a negative interaction for yield under field conditions has been reported [115]. In such cases, the advantage of genome editing tools is their ability to edit some complex traits that are not performing phenotypically as per expectation. Genome editing through the CRISPR/Cas9 strategy can introduce the desirable traits into a genetic background in a very short period of time, without crossing and back-crossing procedures [116].

## 7. Adoption

The adoption of these technologies and practices, complemented by better infrastructure and market access, will increase the income and livelihood of farmers and the poor. However, ironically, despite a big push from suppliers and high demand from consumers, the adoption of many of these technologies and practices is low because of problems related to technology, institutions and policy. The main challenges to large-scale adoption of DSR technologies include a lack of farmers' knowledge about the available technologies, farmers' mindset on the traditional cultivational practices, poor linkages among the stakeholders and insufficient government support. Regulatory reform, constant support of research and extension agencies and public infrastructure investments can provide the needed support to the farmers to ensure the real scaling up of DSR technologies in Asian countries. Precision farming involving the extensive uses of information and communication technologies, remote sensing using the satellite technologies, geographical information systems (GIS), soil and agronomy sciences may bring a breakthrough revolution in the world of E-agriculture. Large-scale dissemination and adoption of these high-level technologies for rice-based systems can sustainably increase rice production, improve food security, reduce poverty and accelerate rural transformation.

## 8. Conclusions

Climate change-induced major abiotic stresses, such as flood, drought, salinity, cold and high temperature, are considered as notable threats to rice production and causes of significant yield loss. DSR has emerged as an efficient, economically viable and environmentally promising alternative to PTR with lower water-labor use and production costs. An

ongoing large-scale shift towards DSR necessitates a convergence of breeding, agronomic and other approaches for its sustenance and harnessing natural resources and environmental benefits. It is very important to take note on the right combinations of suitable DSR varieties and technologies that allow farmers to realize the significant economic returns from DSR cultivation with less water-labor energy and nutrients. The agro-ecological and socio-economic changes in the Asian countries call not only for the development but also for the dissemination and proper adoption of high-yielding DSR-adapted rice varieties and technologies utilizing natural resources. Multiple stress tolerance DSR-adapted rice varieties must be considered in breeding for long-term adaptability to adverse environments. New breeding tools and techniques, such as marker-assisted breeding, pyramiding, genome-wide association studies, haplotype breeding, genomic selection and genome editing, are emerging rapidly from advances in genomic research for the application in rice crop improvement. These innovative breeding methods reduce the breeding cycle time and allow more genetic gain to overcome climate change and other problems in rice production. Furthermore, the supporting technologies, including precise laser land levelling, mechanized seeding, proper water-nutrient-weed management and mechanized harvesting and threshing, might enable the successful and effective expansion and adoption of suitable DSAR varieties and technologies on a large scale.

**Author Contributions:** Conceptualization, A.K., V.K.S., N.S.; writing—original draft preparation, N.S. and S.Y.; writing—review and editing, A.K. and V.K.S.; supervision, A.K.; funding acquisition, A.K., V.K.S., and N.S. All authors have read and agreed to the published version of the manuscript.

**Funding:** The authors thank the Department of Biotechnology, India (grant no: BT/PR31462/ATGC/127/6/2019) for providing financial support to complete the study.

**Data Availability Statement:** All the supporting data have been provided with the review.

**Conflicts of Interest:** The authors declare no conflict of interest.

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
