# Peer review of "Effective Crop Management and Modern Breeding Strategies to Ensure Higher Crop Productivity under Direct Seeded Rice Cultivation System: A Review"

_agronomy, doi:10.3390/agronomy11071264_

Round 1
Reviewer 1 Report
The manuscript untitled “Effective crop management and modern breeding strategies to ensure higher crop productivity under direct seeded rice cultivation system” addresses an important topic that promotes resource conservation technology and reduces water and labor use at 50%.
Overall the manuscript is well done, with minor grammar and spelling English needed by an English-native speaker, with an important topic to be considered at Agronomy Journal. Also, I would recommend the numbering of sections to easily follow each section approached in the manuscript.
Despite the great amount of work I believe the manuscript still needs major revisions, namely:
L38- data from FAOSTAT in a graphical manner would be important, as also Africa is an important continent contributing to both rice supply and consumption.
L110- Considering this is a review paper, a schematic representation of the DSR technology would be valuable.
L124- At the Water and labor use section, it seems to be presented a case-study in Cambodja. Do you have more data on other places, namely in Asia versus Africa? Considering this is a review it could be important to make such comparison if possible. Instead, just refer more objectively the Cambodja data as a case-study.
Table 4- Amount are in US dollars? Please include currency.
L329- In the section “Grain yield and DSR adapted rice varieties, it would be important to present major rice varieties (its availabilities at markets or ex-situ germplasm collections) and grain yield in the table form with references for a more easily readership of the data.
At table 7, please include references in a new column.
L551- “replace “Till”- by “Until”.
Author Response
Reviewer’s comment
The manuscript untitled “Effective crop management and modern breeding strategies to ensure higher crop productivity under direct seeded rice cultivation system” addresses an important topic that promotes resource conservation technology and reduces water and labor use at 50%. Overall, the manuscript is well done, with minor grammar and spelling English needed by an English-native speaker, with an important topic to be considered at Agronomy Journal. Also, I would recommend the numbering of sections to easily follow each section approached in the manuscript.
Response to reviewer’s comment:
As per reviewer’s comment, the grammar, spelling English language of the manuscript have been revised. The numbering of section and subsections has been done in the revised version of the manuscript. Please see the revised version of the manuscript
Reviewer’s comment
Despite the great amount of work I believe the manuscript still needs major revisions, namely:
L38- data from FAOSTAT in a graphical manner would be important, as also Africa is an important continent contributing to both rice supply and consumption.
Response to reviewer’s comment: As per reviewer’s comment the data from FAOSTAT in a graphical manner has been added in the revised version of the manuscript. Please see lines 38 to 40 and Figure 1 in the revised version of the manuscript.
Reviewer’s comment
L110- Considering this is a review paper, a schematic representation of the DSR technology would be valuable.
Response to reviewer’s comment: As per reviewer’s comment, the schematic representation of the steps involved in the development and release of direct seeded adapted rice varieties is presented in figure 5. Please see figure 5 in the revised version of the manuscript and lines 498-500 in the text.
Reviewer’s comment
L124- At the Water and labor use section, it seems to be presented a case-study in Cambodja. Do you have more data on other places, namely in Asia versus Africa? Considering this is a review it could be important to make such comparison if possible. Instead, just refer more objectively the Cambodia data as a case-study.
Response to reviewer’s comment: We do have data available from Cambodia location only. The data from other studies have already been included in the subsection 2.1. “Water and labor use.” Please see subsection 2.1 in the revised version of the manuscript. As per reviewer’s comment the examples from Cambodia, Nepal and Punjab have been added under subsection 2.1.1 in the revised version of the manuscript. Please see lines 148-162 in the revised version of the manuscript.
Reviewer’s comment
Table 4- Amount are in US dollars? Please include currency.
Response to reviewer’s comment: The cost is in NRs: Nepalese Rupee. As per reviewer’s comment the currency has been added in the table 4. Please see Table 4 in the revised version of the manuscript.
Reviewer’s comment
L329- In the section “Grain yield and DSR adapted rice varieties, it would be important to present major rice varieties (its availabilities at markets or ex-situ germplasm collections) and grain yield in the table form with references for a more easily readership of the data.
Response to reviewer’s comment: The detailed information on the varieties and grain yield have already been added in the Table 5 of the revised manuscript. Please see Table 5.
Reviewer’s comment
At table 7, please include references in a new column.
Response to reviewer’s comment: The data in the Table 7 is from genomics-assisted breeding study conducted by Sandhu et al. 2021 at IRRI. The manuscript has now been accepted for publication in Frontiers in Plant Science journal. The source has been added in the figure legend. Please see table 7 in the revised version of the manuscript.
Reviewer’s comment
L551- “replace “Till”- by “Until”.
Response to reviewer’s comment: As per reviewer’s comment, till has been replaced by until. Please see line 542 in the revised version of the manuscript.
Reviewer 2 Report
Rice is the most important crop in many regions of the world. There are no reports in the literature which comprehensively present the available information on its management and breeding strategies. This underlines the importance of the manuscript.
It is not clear from the title or abstract that this is a literature review.
Author Response
Reviewer’s comment
Rice is the most important crop in many regions of the world. There are no reports in the literature which comprehensively present the available information on its management and breeding strategies. This underlines the importance of the manuscript.
Response to reviewer’s comment: Thank you for the comment.
Reviewer’s comment
It is not clear from the title or abstract that this is a literature review.
Response to reviewer’s comment: As per reviewer’s comment, review word has been added in the title as “Effective crop management and modern breeding strategies to ensure higher crop productivity under direct seeded rice cultivation system: A review” and in the abstract as “The review provides information on the traits, donors, genes/QTL needed for DSR and the available DSR adapted breeding lines. Further, the information’s are supplemented with a discussion on constrains and needed policies in scaling up the DSR adoption.” Please see the title and lines 22-25 in the revised version of the manuscript.
Reviewer 3 Report
The manuscript is written in a clear way, easy to read and with appropriate reference to data in figures and tables.
Author Response
Reviewer’s comment
The manuscript is written in a clear way, easy to read and with appropriate reference to data in figures and tables.
Response to reviewer’s comment: Thank you for the comment.
Round 2
Reviewer 1 Report
The authors have improved signficantly the manuscript by addressing all issues commented at an earlier version.
I believe that the ms can now be accepted for publication.
Only a minor, but an important amendment, at Table 4 currency NT (Nepalese Rupee) can be used but include US$, for a wider value comprehension.